# Sample-Efficient Differentially Private Fine-Tuning via Gradient Matrix Denoising

## Abstract

We address the challenge of sample efficiency in differentially private fine-tuning of large language models (LLMs) using DP-SGD. While DP-SGD provides strong privacy guarantees, the added noise significantly increases the entropy of gradient matrices, disrupting their low-rank structure and slowing optimization. We propose a post-processing algorithm that leverages random matrix theory to denoise gradients, restore low-rank structure, and improve alignment with the original signal. Applied to DP-SGD fine-tuning of RoBERTa model family on GLUE tasks and Qwen and Llama family on DART and E2E datasets, our method improves sample efficiency compared to state-of-the-art approaches, substantially reducing training time when optimal performance is not required. This work demonstrates that matrix recovery techniques can enhance the utility of private language model training without compromising privacy guarantees.

## 1 Introduction

Many applications of machine learning in natural language processing tasks may raise privacy concerns, because of the potential data leakage from using models trained on private data (Carlini et al., 2021; 2022). Differential privacy (DP) (Dwork et al., 2014) is a formal framework for quantifying and limiting the privacy loss experienced by individuals whose data are included in a dataset when an algorithm is applied to it. DP-SGD (Abadi et al., 2016), is a method to ensure privacy guarantees as measured by the DP framework, and has been succesfully applied to NLP tasks (Yu et al., 2021; Li et al., 2021).

Applying DP-SGD to language models, while successful, has many challenges. Training large language models (LLMs) with DP-SGD is computationally expensive (Li et al., 2021). Using parameter efficient fine-tuning methods, this challenge has been addressed (Yu et al., 2021). Still, computational cost is higher than the non-private training, because of lower sample efficiency. This can be viwed, for example, in Figure 1.

In the DP-SGD method, noise is deliberately added to the gradient vector before it is passed to the optimizer to ensure privacy. While this step is crucial for protecting individual data, it also complicates the optimization process. Specifically, the added noise alters the distribution of singular values in the gradient matrix. For transformer-based language models, the singular values of the gradient matrix typically decay rapidly, reflecting low matrix entropy and a strong low-rank structure (Li et al., 2022; Zhao et al., 2024). After noise is introduced, however, the singular values decay more slowly, leading to higher matrix entropy (Li et al., 2022). We hypothesize that this increase in entropy makes optimization more difficult.

The singular values of the gradient matrix undergo a "phase transition" (Baik et al., 2005) when noise is added. If the underlying signal is weak, the singular values of the noisy matrix become indistinguishable from those of pure noise. Figure 4 illustrates this by comparing the sorted singular values of a RoBERTa layer's gradient matrix before and after DP-SGD noise is applied. In this weak-signal regime, the noisy gradient's singular values closely follow the "bulk" distribution predicted by the Marchenko–Pastur law (Marčenko & Pastur, 1967; Tao, 2012), making them essentially indistinguishable from pure noise. Thus, when a low-rank signal is too small relative to the noise, it is hidden in the noise and cannot be detected or recovered by examining the singular values and vectors alone. This highlights a fundamental limitation: sufficiently weak signals are undetectable in the presence of strong noise.

However, if some singular values exceed this threshold, the largest singular values of the noisy matrix deviate from the bulk, as shown in Figure 5. This phenomenon is known as the Baik–Ben Arous–Péché (BBP) phase transition (Baik et al., 2005). The extent of these deviations, as well as the alignment between the singular vectors of the noisy and original matrices, can be predicted mathematically (Baik & Silverstein, 2006; Benaych-Georges & Nadakuditi, 2012). These properties enable partial recovery of the original matrix from its noisy observation (Shabalin & Nobel, 2013; Gavish & Donoho, 2014).

In this paper, we propose a post-processing algorithm for DP-SGD that leverages the mentioned matrix recovery techniques from random matrix theory to reduce the entropy of the gradient matrix, restore its low-rank structure, and improve the alignment between the noisy and original gradients. To evaluate our approach, we apply it to DP-SGD fine-tuning of RoBERTa (Liu et al., 2019) on GLUE tasks (Wang et al., 2019). We compare the sample efficiency of our method to the current state-of-the-art (Yu et al., 2021), demonstrating that our approach can improve the sample efficiency of DP-SGD fine-tuning for language models. While our method may not always achieve the highest possible utility, it can substantially reduce training time when optimal performance is not required.

## 2 PRELIMINARIES

### 2.1 DIFFERENTIAL PRIVACY

Differential privacy is a framework to quantify and measure the maximum possible privacy risks an algorithm with sensitive training dataset may have. For a pair $(\epsilon, \delta)$, this formalism asks any learning algorithm $\mathcal{M}$ to have similar outputs for two datasets differing only in one element. Intuitively, the output of the learning algorithm should not change much whether it sees a particular example or not. This intuition can be formulated mathematically in the concept of approximate differential privacy.

#### 2.1.1 APPROXIMATE DIFFERENTIAL PRIVACY

**Definition 1.** *Two sets are called neighboring sets if they differ only in inclusion or exclusion of exactly one element.*

**Definition 2.** *A randomized algorithm $\mathcal{M}$ is said to satisfy $(\epsilon, \delta)$ differential privacy, if for any two neighboring datasets $D$ and $D'$ and for any event $E$, the following holds*

$$\mathbb{P}(\mathcal{M}(D) \in E) \leq \exp(\epsilon)\mathbb{P}(\mathcal{M}(D') \in E) + \delta \tag{1}$$

.

*In practice, it is usual to have $\delta$ in the order of $|D|^{-1}$ (Abadi et al., 2016). In NLP applications, $\epsilon$ usually takes values between $0.5$ and $8$ (Yu et al., 2021; Li et al., 2021).*

#### 2.1.2 DP-SGD

DP-SGD is a popular method of training deep learning models with approximate differential privacy guarantees. This method is a modification of the popular first order SGD algorithm.

DP-SGD works by modifying the gradient before passing it to the optimizer. It has two main parts 1. per example gradient clipping and 2. noise addition. There are two hyper-parameters associated with each of them, the clipping threshold, $C$, which controls the maximum per example gradient norm, and the noise multiplier, $\sigma$, which when multiplied by $C$, controls the standard deviation of the isotropic zero mean Gaussian noise added to the sum of the clipped gradients (Abadi et al., 2016).

In standard SGD, for a batch of data $\{x_i\}_{i \in \mathcal{B}} \subset D$, the batch gradient is computed as:

$$\boldsymbol{g}_\mathcal{B} = \frac{1}{B} \sum_{i \in \mathcal{B}} \boldsymbol{g}_i = \frac{1}{B} \sum_{i \in \mathcal{B}} \nabla f(\theta, x_i)$$

In DP-SGD, each individual gradient is first clipped so that its norm does not exceed the threshold $C$. The clipped gradients are then summed, and Gaussian noise with entries drawn from $\mathcal{N}(0, \sigma^2 C^2)$ is added. Finally, the result is averaged over the batch:

$$\bar{g}_{\mathcal{B}} = \sum_{i \in \mathcal{B}} \mathrm{clip}(g_i, C)$$

$$\tilde{g}_{\mathcal{B}} = \frac{1}{B} \left( \bar{g}_{\mathcal{B}} + w \right), \quad w_j \sim \mathcal{N}(0, \sigma^2 C^2) \tag{2}$$

This new gradient will then be fed to the optimizing algorithm of the choice, e.g. SGD or Adam(W). While $\sigma$ and $C$ are hyper-parameters, the constant $\sigma$ is selected based on the privacy guarantees desired for the model $(\epsilon, \delta)$, the number of training steps, sampling rate $(\frac{B}{|D|})$. The method for computing the necassiry $\sigma$ based on the privacy gaurantees is called the privacy accountant. For this work, we use the privacy accountant of Gopi et al. (2021) which currently is the most tight privacy accountant.

---

**Algorithm 1** DP-SGD (with Denoising)

---

**Require:** Dataset $D$, loss function $f(\theta, x)$, model parameters $\theta$, sampling rate $\rho$, clipping norm $C$, noise multiplier $\sigma$, optimizer $\mathcal{O}$, number of steps $T$, Denoising function $\mathrm{Denoise}(\cdot)$
  **for** $t \leftarrow 1$ to $T$ **do**
    Sample a batch $\mathcal{B}$ from $D$ using Poisson sampling with rate $\rho$
    **for** each $i \in \mathcal{B}$ **do**
      Compute per-example gradient $g_i \leftarrow \nabla_\theta f(\theta, x_i)$
      Clip gradient: $\mathrm{clip}(g_i, C) \leftarrow g_i / \max(1, \|g_i\|_2 / C)$
    **end for**
    Aggregate clipped gradients: $\bar{g} \leftarrow \sum_{i \in \mathcal{B}} \mathrm{clip}(g_i, C)$
    Draw noise vector $w$ with i.i.d. entries from $\mathcal{N}(0, \sigma^2 C^2)$
    Compute noisy average: $\tilde{g} \leftarrow (\bar{g} + w)/|\mathcal{B}|$
    **Denoise the gradient:** $\hat{g} \leftarrow \mathrm{Denoise}(\tilde{g})$
    Update optimizer state: $\mathcal{O} \leftarrow \mathrm{UpdateState}(\mathcal{O}, \hat{g})$
    Update parameters: $\theta \leftarrow \mathrm{UpdateParameters}(\theta, \mathcal{O})$
  **end for**

---

### 2.1.3 POST PROCESSING INVARIANCE

A fundamental property of differential privacy is its invariance under post-processing. This means that no adversary, regardless of the method applied to the output of a differentially private algorithm, can reduce its privacy guarantees or extract more information about the original dataset. In other words, post-processing cannot make the output less private, providing strong protection against attempts to compromise privacy. While previous work has leveraged this property to improve the utility of the DP-SGD algorithm (Zhang et al., 2024; Balle & Wang, 2018), none have utilized results from random matrix theory for the post-processing function. To our knowledge, this is the first work to apply such results in the context of DP-SGD.

### 2.2 SINGULAR VALUE DISTRIBUTION OF GRADIENTS

The gradients of linear layers of neural networks in training, when viewed as a linear operator, exhibit a low rank structure (Li et al., 2022), (Zhao et al., 2024). Viewing the singular values of the gradient operator, this translates to a rapid decay in the singular values of the gradient matrix. This is a well known phenomenon in the literature, and has been observed in many different settings, e.g. (Li et al., 2022), (Zhao et al., 2024). While this has been used to explain why differential privacy works so well in deep models with large parameter counts contrary to theoretical expectations (Li et al., 2022), it has not been used to improve the sample efficiency of differentially private training. In this work, we use this property to improve the sample efficiency of differentially private training by using low rank matrix estimation techniques to denoise the gradients before passing them to the optimizer.

## 2.3 LOW RANK MATRIX ESTIMATIONS

Low rank matrix reconstruction is a rich sub-field of signal processing (Donoho et al., 2018; Gavish & Donoho, 2014; Shabalin & Nobel, 2013). Assuming the rank of the signal matrix $\boldsymbol{X} \in \mathbb{R}^{m \times n}$ is $k$, we can use the SVD decomposition to write it as

$$\boldsymbol{X} = \sum_{i=1}^{k} \lambda_i \boldsymbol{u}_i \boldsymbol{v}_i^T$$

where $\lambda_i$s are non-increasing singular values, and $\boldsymbol{u}_i \in \mathbb{R}^m$, $\boldsymbol{v}_i \in \mathbb{R}^n$ are orthonormal vectors.

Then, a noise matrix with entries drawn from $\mathcal{N}(0, \sigma^2)$ is added to get the noisy matrix $\tilde{\boldsymbol{X}}$:

$$\tilde{\boldsymbol{X}} = \boldsymbol{X} + \boldsymbol{\Delta}, \quad \boldsymbol{\Delta}_{ij} \stackrel{\text{i.i.d.}}{\sim} \mathcal{N}(0, \sigma^2)$$

The goal is to estimate the original matrix $\boldsymbol{X}$ from the noisy observation $\tilde{\boldsymbol{X}}$. We write the SVD decomposition of $\tilde{\boldsymbol{X}}$ in the notation

$$\tilde{\boldsymbol{X}} = \sum_{i=1}^{\min(m,n)} \tilde{\lambda}_i \tilde{\boldsymbol{u}}_i \tilde{\boldsymbol{v}}_i^T$$

Note that the noisy version may (and usulaly does) have more than $k$ non-zero components.

### 2.3.1 EFFECT OF NOISE ON SINGULAR VALUES AND SINGULAR VECTORS: A PHASE TRANSITION

With the mentioned notation we have

$$\tilde{\lambda}_i \approx \begin{cases} F_{\sigma,n,m}(\lambda_i) = \sqrt{\left(\lambda_i + \frac{\sigma^2 n}{\lambda_i}\right)\left(\lambda_i + \frac{\sigma^2 m}{\lambda_i}\right)} & \text{if } \lambda_i > \sigma \sqrt[4]{mn} \\ \sigma(\sqrt{m} + \sqrt{n}) & \text{if } \lambda_i \le \sigma \sqrt[4]{mn} \end{cases} \tag{3}$$

This is an increase in the value of the singular value, which is usual in random matrix theory. It is important to note that these results are typically stated in the asymptotic regime, where the matrix dimensions grow to infinity and the noise variance may scale with the dimensions, often under specific assumptions on the ratio $m/n$. In practical, finite-dimensional settings, these approximations may incur some error. The precise rate of this error in finite dimensions is not addressed here and could be an interesting direction for further study. The derivation of these results from their asymptotic forms is postponed to the appendix A.2.

Also, assuming all of the eignvalues of $\boldsymbol{X}$ are distinct, and if $\lambda_i > \sigma \sqrt[4]{mn}$, following Lemma 3 of Gavish & Donoho (2014) or proposition 9 of Shabalin & Nobel (2013), we can write

$$|\langle \boldsymbol{u}_i, \tilde{\boldsymbol{u}}_j \rangle|^2 \approx \begin{cases} \dfrac{\lambda_i^4 - mn\sigma^4}{\lambda_i^4 + m\lambda_i^2 \sigma^2} & i = j \\ 0 & i \ne j \end{cases}, \tag{4}$$

and

$$|\langle \boldsymbol{v}_i, \tilde{\boldsymbol{v}}_j \rangle|^2 \approx \begin{cases} \dfrac{\lambda_i^4 - mn\sigma^4}{\lambda_i^4 + n\lambda_i^2 \sigma^2} & i = j \\ 0 & i \ne j \end{cases}. \tag{5}$$

However if $\lambda_i \le \sigma \sqrt[4]{mn}$, then

$$|\langle \boldsymbol{u}_i, \tilde{\boldsymbol{u}}_j \rangle|^2 \approx |\langle \boldsymbol{v}_i, \tilde{\boldsymbol{v}}_j \rangle|^2 \approx 0 \tag{6}$$

### 2.3.2 MATRIX DENOISING

Matrix denoising methods aim to recover the underlying signal matrix $\boldsymbol{X}$ from its noisy observation $\tilde{\boldsymbol{X}}$ by leveraging the low-rank structure of the signal. Many of these methods shrink the singular values of the noisy matrix. One such approach is the so-called optimal method discussed in Shabalin & Nobel (2013); Donoho et al. (2018), which outputs a low-rank matrix.

**Optimal Denoising** The mentioned optimal estimator for the signal matrix can be written as

$$\hat{\boldsymbol{X}}_{\text{optimal}} = \sum_{i=1}^{r} \eta_i \tilde{\boldsymbol{u}}_i \tilde{\boldsymbol{v}}_i^T \tag{7}$$

where, following Shabalin & Nobel (2013), the optimal coefficients are

$$\eta_i = \hat{\lambda}_i \cdot \sqrt{\frac{\hat{\lambda}_i^4 - mn\sigma^4}{\hat{\lambda}_i^4 + m\hat{\lambda}_i^2\sigma^2}} \cdot \sqrt{\frac{\hat{\lambda}_i^4 - mn\sigma^4}{\hat{\lambda}_i^4 + n\hat{\lambda}_i^2\sigma^2}}, \quad \text{where} \quad \hat{\lambda}_i = F_{\sigma,n,m}^{-1}(\tilde{\lambda}_i) \tag{8}$$

for all $i$ such that $\tilde{\lambda}_i > \sigma(\sqrt{m} + \sqrt{n})$, and zero otherwise. It has been shown to achieve the best possible mean squared error (MSE) under certain conditions, particularly when the noise is Gaussian and the signal is low-rank.

**Computational Overhead** Although the additional computation required for matrix denoising may seem significant, with careful implmenetation and also utilizing the parallelizable nature of the optimal denoising algorithm, the overhead can be kept minimal. We draw readers attention to three properties that can be leveraged to reduce the computational overhead. 1) First, the fact that in equation 7, the $F_{\sigma,n,m}^{-1}(\tilde{\lambda}_i)$ can be computed independently for each singular value, and the same applies to the computation of $\eta_i$. 2) Second, the SVD computation, which is the most computationally expensive part of the algorithm, can be computed in a batched manner for all the layers of the neural network with similar dimensions. And 3) Third, by utilizing approximate methods when acceptable, the SVD computation can be further accelerated. While if implmeneted naïvely, the overhead can be just short of $3\%$ of the total training time, combining the mentioned strategies, the computational overhead of the denoising step can be reduced to less than $1\%$ of the total training time in our experiments (Table 1). The efficient implementation is available as part of the supplementary material.

Table 1: Training time in seconds for different methods of DP-SGD fine-tuning of RoBERTa on SST-2 dataset with the setting of section 4. The overhead is computed as the percentage increase in training time compared to regular DP-SGD.

| Model | Method | Train Time | Overhead |
|-------|--------|------------|----------|
| RoBERTa Base | Regular DP-SGD | 1846 | – |
| | Naïve denoising implementation | 1899 | 2.87% |
| | Efficient denoising implementation | 1861 | 0.81% |
| RoBERTa Large | Regular DP-SGD | 5658 | – |
| | Naïve denoising implementation | 5781 | 2.17% |
| | Efficient denoising implementation | 5700 | 0.74% |

## 3 METHODOLOGY

In this section, we introduce our post-processing method, which leverages equations 7 and 8 to denoise the gradients produced by DP-SGD before they are passed to the optimizer. First, we establish that the slower convergence observed with DP-SGD (compared to non-private training) is primarily caused by the added Gaussian noise, not by gradient clipping. Although prior work often attributes

the loss gap between DP-SGD and non-private training to clipping (Bu et al., 2023), these are distinct phenomena. Figure 1 shows that the final validation loss is similar for DP-SGD with a zero noise multiplier ($\sigma = 0$) and for DP-SGD using the noise level required for privacy; however, convergence toward that final loss is significantly slower when noise is added. This indicates that the added noise is the main driver of DP-SGD's slower convergence. This is why we focus on denoising the gradients in our method. More similar experiments comparing clipping alone versus DP-SGD can be found in appendix B.

Figure 1: Comparison of validation loss between DP-SGD and training using per-sample clipping only (no noise injected). This is for training RoBERTa base model on sst-2 dataset. The Setting are similar to what is described in 4, except that for only clipping method, we do not add the gaussian noise and only perform the per-sample clipping. It is evident that the slowdown in convergence is more pronounced for DP-SGD than for clipping alone.

### 3.1 FRAMEWORK

We apply the denoising method by aiming to increase the alignment between the denoised gradient and the clipped gradient. Specifically, our objective is to construct a denoising function that, given the noisy gradient as input, produces an output that is more closely aligned with the clipped gradient. Using the notation from Section 2.1.2, we seek a denoising function $\mathrm{Denoise}(\cdot)$ such that

$$\cos(\mathrm{Denoise}(\tilde{\boldsymbol{g}}), \bar{\boldsymbol{g}}) > \cos(\tilde{\boldsymbol{g}}, \bar{\boldsymbol{g}})$$

where $\cos(\boldsymbol{a}, \boldsymbol{b}) = \frac{\boldsymbol{a}^T \boldsymbol{b}}{\|\boldsymbol{a}\|_2 \|\boldsymbol{b}\|_2}$ is the cosine similarity between two vectors $\boldsymbol{a}$ and $\boldsymbol{b}$.

For tracking this value for evaluation purposes, we define the Improvement at step $t$ as

$$\mathrm{Improvement}(t) = \cos(\mathrm{Denoise}(\tilde{\boldsymbol{g}}_t), \bar{\boldsymbol{g}}_t) - \cos(\tilde{\boldsymbol{g}}_t, \bar{\boldsymbol{g}}_t)$$

If we can come up with such a denoising function, we hope to improve the sample efficiency of DP-SGD by making the noisy gradients more closely resemble the true (clipped) gradients. Having such a denoising function, we can change the DP-SGD algorithm as in Algorithm **??**.

We expect that if the improvement at each step $t$ is consistently non negative, $\mathrm{Improvement}(t) \geq 0$, then the denoising function is effectively aligning the noisy gradients with the true (clipped) gradients, leading to faster convergence of the DP-SGD algorithm. It is important to note that the improvement function is only used for evaluation porpuses and is not used for the algorithm itself, as doing so would violate the differential privacy criteria. The following sections will detail the implementation of the denoising function which are mainly based on the results reviewed in Section 2.3.2. The guiding principle behind adapting the matrix denoising methods to our task is that the denoiser should increase alignment with the (private) clipped gradient, while itself using no additional private information. This principle helps us in two design choices. One gives us a way to adapt the asymptotic formulas to finite dimension, and the other helps us to generalize the denoising algorithm from operating on a single matrix to operating on a collection of matrices (the layers of the neural network).

## 3.2 DENOISING FUNCTION

The denoising function we propose is basically application of the denoising functions in section 2.3.2 to the linear components of the noisy gradient $\tilde{g}$. Supposing $W$ is a layer of our neural network $\theta$, the restriction of the (clipped) gradient to $W$ is a matrix $\sum_{x \in \mathcal{B}} \text{clip}(\nabla_\theta f(\theta, x), C)|_{W} = \bar{g}|_{W}$. If we consider all the different layers of the neural network, the parameters of the neural network can be partitioned as

$$\theta = W_1 \times W_2 \times \ldots \times W_L$$

where $L$ is the number of layers in the network. Then, we can write

$$\bar{g} = (\bar{g}|_{W_1}, \bar{g}|_{W_2}, \ldots, \bar{g}|_{W_L})$$
$$\tilde{g} = (\tilde{g}|_{W_1}, \tilde{g}|_{W_2}, \ldots, \tilde{g}|_{W_L})$$

With this notation, we can define the denoising function as seperate application of the denoising functions to each layer's gradient:

$$\text{Denoise}(\tilde{g}) = (\text{Denoise}(\tilde{g}|_{W_1}), \text{Denoise}(\tilde{g}|_{W_2}), \ldots, \text{Denoise}(\tilde{g}|_{W_L}))$$

Where if a layer $W_i$ is not a linear layer, we simply set $\text{Denoise}(\tilde{g}_{W_i}) = \tilde{g}_{W_i}$. With this notation, we can also define the per-layer improvement as

$$\text{Improvement}_i(t) = \cos(\text{Denoise}(\tilde{g}_t|_{W_i}), \bar{g}_t|_{W_i}) - \cos(\tilde{g}_t|_{W_i}, \bar{g}_t|_{W_i}).$$

For linear layers, we modify the so called "optimal" denoising method in two ways.

- **To adapt the assymptotic formulas to finite dimension** we only apply optimal denoising if the singular values of the noisy layer gradient are larger than a preset multiple of $\sigma(\sqrt{n} + \sqrt{m})$, where $n, m$ are dimensions of that linear layer. When the singular value is larger than the required threshold, we apply the optimal denoising function.
- **To keep the gradient norm the same and also making sure the per-layer alignment improvement will result in whole graident improvement**, we rescale the denoised per-layer graidnet so that its $\ell_2$ norm is equal to the noisy version $\tilde{g} \mapsto \frac{||\tilde{g}||}{||\hat{g}_{\text{optimal}}||}\hat{g}_{\text{optimal}}$.

So for linear layers and the hyperparameter $\kappa$ we have

$$\text{Denoise}(\tilde{g}|_{W}) = \begin{cases} \tilde{g}|_{W}, & \text{if } \lambda_1(\tilde{g}|_{W}) < \kappa\,\sigma(\sqrt{n} + \sqrt{m}) \\ \frac{||\tilde{g}||}{||\hat{g}_{\text{optimal}}||}\hat{g}_{\text{optimal}}, & \text{otherwise} \end{cases} \quad (9)$$

### 3.2.1 WHY THRESHOLD IS NEEDED AND WHY THIS SPECIFIC VALUE?

It is important to recognize that the results in Section 2.3.2 are derived in asymptotic settings. For instance, the theory predicts that if all singular values of the signal matrix are less than $\sigma\sqrt[4]{nm}$, or equivalently, if all singular values of the noisy matrix are less than $\sigma(\sqrt{n} + \sqrt{m})$, then the inner products between the left (or right) singular vectors of the signal and noisy matrices should be zero. In that case, the optimal denosing algorithm returns the zero matrix as the optimum result and states that it is the best one can get. However, in practice and for finite-dimensional matrices, this is not true, and the noisy gradient, even if its singular value are small, usually still has some positive cosine similarity with the original gradient. As a result, the denoising algorithm does not always improve the alignment between the noisy and per-layer clipped gradients.

Fortunately, we identified a simple RMT-based criterion to decide when to apply the denoiser. Concretely, we only denoise a layer if the largest singular value of its noisy gradient exceeds $\kappa\,\sigma(\sqrt{n} + \sqrt{m})$. Choosing $\kappa = 1$ prevents denoising in cases where the optimal estimator would return the zero matrix. This threshold is motivated by the phase transition in equation 3: $\sigma(\sqrt{n} + \sqrt{m})$ is the maximal singular value of a pure-noise matrix (signal equal to zero). Thus, RMT indicates when denoising can meaningfully improve alignment; if the estimator would output the zero matrix, the theoretical prediction is that there is no signal to recover.

Our observations show that for correct value of $\kappa$, denoising tends to improve the alignment, otherwise it may reduce the alignment . For choosing the value of $\kappa$, we tuned it on the SST dataset while training the RoBERTa-base model by choosing the best value from the set $\{1.01, 1.02, 1.05, 1.1\}$, and used the same value for all the other model/datset pairs. We choose this set to have values greater than $1$ to avoid the cases where denoising outputs zero matrix. Also, we want to keep the values as close to $1$ as possible to have more layers denoised. The trade-off here is to have a big enough $\kappa$ so the denoising improves the alignment, and to have it small enough so that we get enough per-layer gradients denoised to get the most out of the alignment improvement. The best value we found was $\kappa = 1.02$ (Figure 2). We also like to emphasis that while we could have tuned $\kappa$ for each model/dataset pair, we refrained from doing so to show the robustness of our method to this hyperparameter. The value $\kappa = 1.02$ worked well accross all the model/dataset pairs we tried.

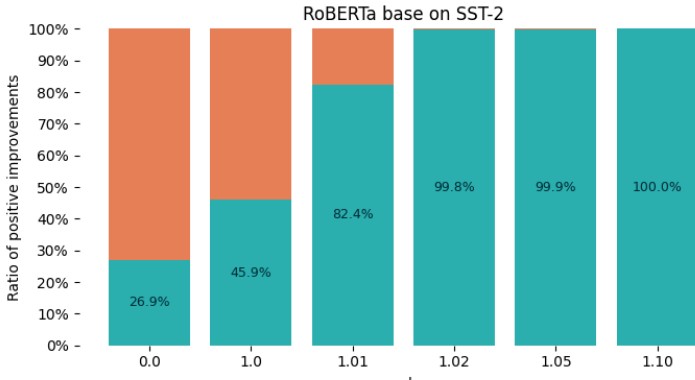

Figure 2: Effect of different values of $\kappa$ on the per-layer improvements when training RoBERTa base model on sst-2 dataset. The Setting are similar to what is described in 4. It is evident that $\kappa = 1.02$ is the smallest $\kappa$ which has dominantly positive per-layer improvement.

### 3.2.2 WHY NORM CORRECTION IS NEEDED?

The scope in which the denoising function from the RMT works is to improve the alignment between the noisy and clipped per-layer gradients. There is no extension of the RMT method to a combination of different layers, and the relative scaling between them. For generalizing to a method for improving alignment of the whole gradient vector, we are going to use the following theorem, which states that if we improve the alignment of each component of a vector, and keep their norms the same, then the overall alignment will also improve.

**Theorem 1.** *Let $x = (x_1, \ldots, x_c) \in \mathbb{R}^n$ be a target vector, with $x_i \in \mathbb{R}^{n_i}$ and $\sum_{i=1}^{c} n_i = n$. Let $y = (y_1, \ldots, y_c), y' = (y'_1, \ldots, y'_c) \in \mathbb{R}^n$ be estimations of $x$, with $y_i, y'_i \in \mathbb{R}^{n_i}$. If we have*

    *(i) $\cos(y_i, x_i) \leq \cos(y'_i, x_i)$, or all $i \in \{1, \ldots, c\}$. and*

    *(ii) $\|y_i\| = \|y'_i\|$, for all $i \in \{1, \ldots, c\}$.*

*Then, we have $\cos(y, x) \leq \cos(y', x)$.*

A proof for this thorem is provided in the appendix D. In our case, the target vector is the clipped gradient $\bar{g}$, and the two estimations are the noisy gradient $\tilde{g}$ and the denoised gradient $\mathrm{Denoise}(\tilde{g})$. It is worth noting that this theorem does not make any assumptions about the nature of the vectors involved, for example if they have any matrix structure at all, and is a general result about cosine similarity and vector norms and is not related to low-rank structure. Also, contrary to the results in section 2.3.2, this theorem is exact and does not rely on any asymptotic approximations or probabilistic arguments.

On another note, the assumption (ii) in theorem 1 is the reason we need to do the norm correction in equation 9. Without this correction, even if the per-layer alignment improves, there is no guarantee that the overall alignment will also improve. To see the effect of the norm correction, we have done

abalation studies for the effect of norm correction. The results for the case of training RoBERTa base model on SST-2 dataset are presented in figure 3. It is evident that without the norm correction, the improvement is not consistent, and even negative. This shows the importance of the norm correction step in our denoising function. More abalation results can be found in the appendix D.

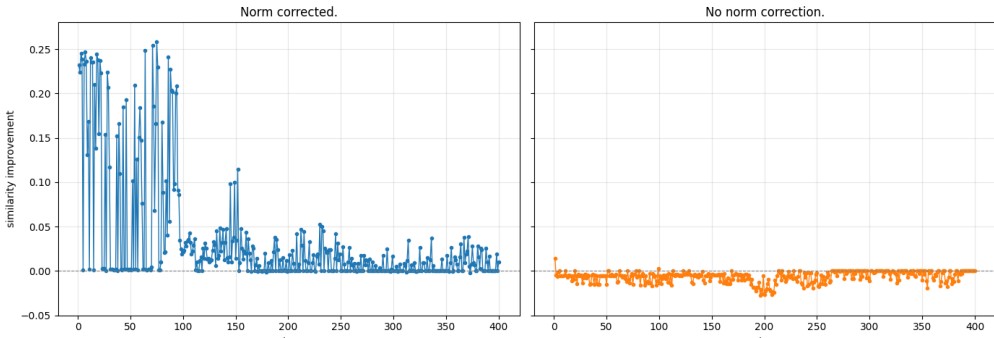

Figure 3: Comparison of whole gradient improvement when using norm correction and not using it. This is for training RoBERTa base model on sst-2 dataset. The Setting are similar to what is described in 4. It is evident that using norm correction results in more consistent improvement.

Also, on the last note, we like to note that the norm of the denosied gradient in this method is exactly the nosie of the noisy gradient. This will prevent any possible issues with convergence of the optimizer due to unexpected changes in the gradient norm.

## 4 EXPERIMENTS

In this section, we present the evaluation method and the experiment results we had. To evaluate our main goal of improving the sample efficiency of DP-SGD, we compared the performance of DP-SGD with and without our denoising method across different datasets from the GLUE benchmark (Wang et al., 2019) and two sizes of the RoBERTa model (Liu et al., 2019).

Because our goal is to find a fast converging method, with possible trade-off in the final performance, we count the number of training steps each method needs to reach some certain (validation) accuracy thresholds. We set these thresholds to be $95\%$ and $90\%$ of the SOTA results for the private training of the same models on the same datasets. The SOTA results are taken from Yu et al. (2021).

For epsilon, we also follow the same setup as Yu et al. (2021), which is $6.7$ for all datasets, and compute the required noise multiplier using the privacy accountant of Gopi et al. (2021) in each case so that the total privacy loss at $400$ steps is $6.7$.

We keep every other hyper-parameter the same as Yu et al. (2021), including batch size, learning rate, weight decay, and clipping norm. Looking at the tables 2 and 3, we can see that our method consistently improves the sample efficiency of DP-SGD across all datasets and model sizes. **Improvements range from 20% to 100%** in the number of steps required to reach $90\%$ and $95\%$ of the SOTA performance. Also, we achieved higher performance in five out of eight cases for the final accuracy at $400$ steps.

We also report additional language-generation experiments on the E2E benchmark (Novikova et al., 2017) and DART (Nan et al., 2021) using Qwen3 and Llama3.2 models (Team, 2025; Grattafiori et al., 2024) (see Appendix E for hyperparameters and more details). After 50 training steps (Table 4), our denoising method outperforms the baseline in **49 of 50** model/dataset/metric combinations, demonstrating substantial sample-efficiency gains under tight iteration budgets. These experiments use models of up to 4B parameters, indicating the approach scales to larger models.

| Task | Method | Final Acc. (at 400 steps) | SOTA (at 20 epochs) | Steps | | Speedup | |
|------|--------|---------------------------|---------------------|-------|-----|---------|-----|
| | | | | 90% | 95% | 90% | 95% |
| SST | Ours | 92.4 | 92.5 | 150 | 150 | 67% | 67% |
| | Baseline | **92.5** | | 250 | 250 | | |
| QNLI | Ours | **84.6** | 87.5 | 200 | 300 | 100% | - |
| | Baseline | 80.0 | | 400 | – | | |
| MNLI | Ours | **80.0** | 83.5 | 250 | 400 | 40% | - |
| | Baseline | 77.6 | | 350 | – | | |
| QQP | Ours | **83.1** | 85.7 | 150 | 250 | 67% | 40% |
| | Baseline | 81.9 | | 250 | 350 | | |

Table 2: Comparison of Ours and Baseline on GLUE tasks when training Roberta Base. Final accuracy, SOTA reference, number of steps needed to reach 90% and 95% of SOTA, and speedups (only for Ours) are reported.

| Task | Method | Final Acc. (at 400 steps) | SOTA (at 20 epochs) | Steps | | Speedup | |
|------|--------|---------------------------|---------------------|-------|-----|---------|-----|
| | | | | 90% | 95% | 90% | 95% |
| SST | Ours | 93.8 | 95.3 | 150 | 150 | 33% | 67% |
| | Baseline | **93.9** | | 200 | 250 | | |
| QNLI | Ours | 88.5 | 90.8 | 150 | 250 | 33% | 20% |
| | Baseline | **89.2** | | 200 | 300 | | |
| MNLI | Ours | **85.6** | 87.8 | 200 | 250 | 25% | 20% |
| | Baseline | 85.3 | | 250 | 300 | | |
| QQP | Ours | **84.7** | 87.4 | 150 | 250 | 33% | 20% |
| | Baseline | 84.1 | | 200 | 300 | | |

Table 3: Comparison of Ours and Baseline on GLUE tasks with RoBERTa Large. Final accuracy, SOTA reference, steps to reach 90% and 95% of SOTA, and speedups (only for Ours) are reported.

## 5 LIMITATIONS AND FUTURE WORK

One major limitation of our method is that it does not always produce the best final performance decpite the fastest convergence. In some experiments, the baseline achieves slightly higher final accuracy than our method. This is particularly puzzling given the consistently positive improvement in cosine similarity between the denoised and noisy gradients relative to the clipped gradients. Further investigation is needed to understand this discrepancy and to identify possible remedies.

## 6 REPRODUCIBILITY STATEMENT

All the necessary code and hyperparameters for reproducing the results in this paper has been made available in the supplementary material.

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

## A RANDOM MATRIX THEORY BACKGROUND

We have included some additional figures and derivations for the random matrix theory results used in this paper.

### A.1 LOW RANK STRUCTURE IN GRADIENTS OF LARGE LANGUAGE MODELS

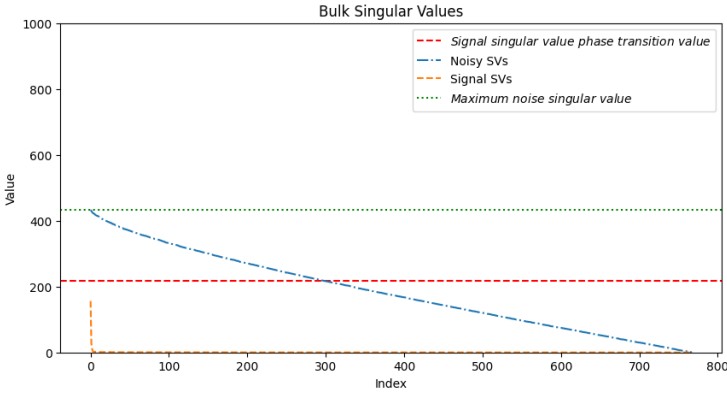

Figure 4: Sorted singular values of the gradient matrix for a RoBERTa layer, before and after adding DP-SGD noise. When the signal singular values are smaller than the red line, the singular values of the noisy matrix are indistinguishable from pure noise.

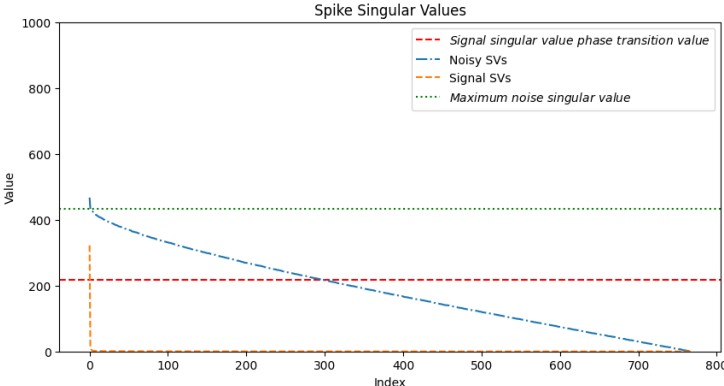

Figure 5: Sorted singular values of the gradient matrix for a RoBERTa layer, before and after adding DP-SGD noise. When some signal singular values exceed the red line, the largest singular values of the noisy matrix deviate from the bulk.

### A.2 FINITE DIMENSIONAL DERIVATION OF RANDOM MATRIX THEORY RESULTS

In the usual random matrix theory literature, the results in Shabalin & Nobel (2013); Donoho et al. (2018); Gavish & Donoho (2014) are stated in the asymptotic regime, where the matrix dimensions grow to infinity and the noise variance may scale with the dimensions. In this section we want to state those results in their original form, and explain the derivation of equations 3, 4, and 5 from their asymptotic forms.

The setup in Shabalin & Nobel (2013); Donoho et al. (2018) is as follows. We have a sequence of matrices $\boldsymbol{X}_n \in \mathbb{R}^{m_n \times n}$ with $m_n/n \to \beta$ as $n \to \infty$. The rank of the signal matrix is fixed, i.e. $\text{rank}(\boldsymbol{X}_n) = r$ for all $n$. The singular values of the signal matrix are fixed, i.e. the non-zero singular values of $\boldsymbol{X}_n$ are $\lambda_1 > \lambda_2 > \ldots > \lambda_r > 0$ for all $n$. Then, we add a noise matrix with i.i.d. entries from $\mathcal{N}(0, 1/n)$ to get the noisy matrix. In these settings, the results in Shabalin & Nobel (2013); Gavish & Donoho (2014) state that

$$
\lim_{n \to \infty} y_{n,i} \stackrel{a.s.}{=}
\begin{cases}
\sqrt{\left(\lambda_i + \dfrac{1}{\lambda_i}\right)\left(\lambda_i + \dfrac{\beta}{\lambda_i}\right)} & \lambda_i > \beta^{1/4} \\
1 + \sqrt{\beta} & \lambda_i \le \beta^{1/4}
\end{cases}
\tag{10}
$$

where $y_{n,i}$ is the $i$-th singular value of the noisy matrix. If we want to change this into the finite dimensional form, we can start form a noise matrix with i.i.d. entries from $\mathcal{N}(0, \sigma^2)$ instead of $\mathcal{N}(0, 1/n)$. Then, if we work with the matrix $\frac{Y}{\sigma\sqrt{n}}$, then, the new noise matrix will have the desired distribution. Using the equation 10 for the matrix $\frac{Y}{\sigma\sqrt{n}}$, and substituting $\beta = m/n$, we get to the equation 3. Similar arguments can be used to derive equations 4 and 5 from their asymptotic forms in Shabalin & Nobel (2013).

## B  WHY DENOISING IS NEEDED?

As discussed in introductory paragraph of section 3, the slowdown in convergence of DP-SGD can be attributed to two main factors: the per-sample gradient clipping and the addition of noise. In this appendix, we present more empirical evidence to support this claim in figure 6.

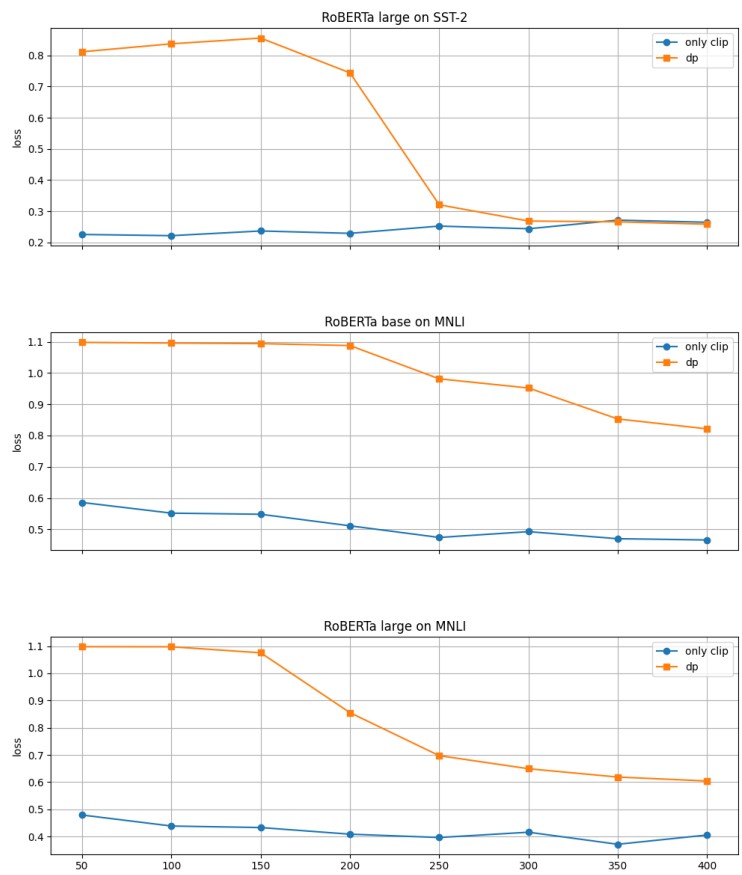

Figure 6: Comparison between validation loss curves of DP-SGD with and without noise addition. The experiments are conducted on the SST-2 and MNLI dataset using the RoBERTa family models. The results clearly indicate that the addition of noise significantly slows down the convergence of the training process compared to the scenario where only gradient clipping is applied. We also see significant gap between the two curves in terms of final loss achieved, for some of the model/dataset pairs .

## C ADDITIONAL RESULTS ON WHY THRESHOLD IS NEEDED.

Here we include additional hyperparameter sweeps for $\kappa$ on different model/dataset pairs. The results are presented in figures 7, 8, and 9. Also, scatter plot of layer improvement vs $\frac{\lambda_1}{\sigma(\sqrt{n}+\sqrt{m})}$ for different layer dimensionality is presented in figure 10 to further justify our choice of threshold.

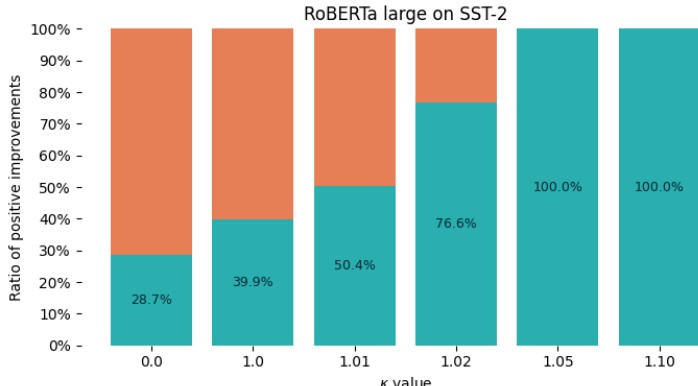

Figure 7: Effect of different values of $\kappa$ on the per-layer improvements when training RoBERTa large model on SST-2 dataset. The Setting are similar to what is described in 4.

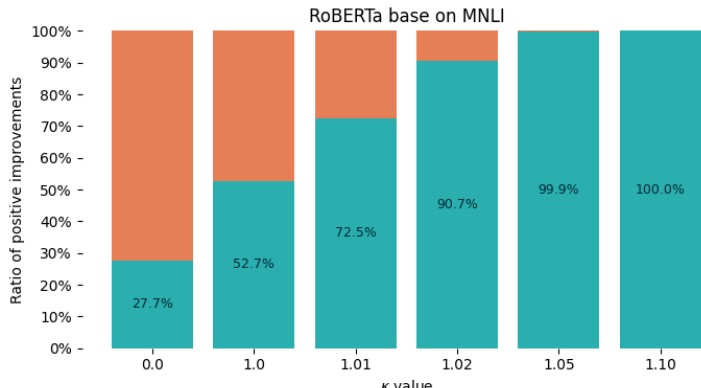

Figure 8: Effect of different values of $\kappa$ on the per-layer improvements when training RoBERTa base model on MNLI dataset. The Setting are similar to what is described in 4.

## D   ADDITIONAL RESULTS ON WHY NORM CORRECTION IS NEEDED.

In this appendix, we give a proof of theorem 1, as well as additional experimental results showing the abalation of norm correction in our denoising function.

### D.1   PROOF OF THEOREM 1

*Proof.* Since the blocks are disjoint and $\|y_i\| = \|y_i'\|$ for all $i$, we can write

$$\langle y, x \rangle = \sum_{i=1}^{c} \langle y_i, x_i \rangle \quad \text{and} \quad \|y\|^2 = \sum_{i=1}^{c} \|y_i\|^2 = \sum_{i=1}^{c} \|y_i'\|^2 = \|y'\|^2.$$

For each block, cosine similarity is

$$\cos(y_i, x_i) = \frac{\langle y_i, x_i \rangle}{\|y_i\| \|x_i\|}.$$

Assumption (i) and the norm equality (ii) imply

$$\langle y_i, x_i \rangle = \|y_i\| \|x_i\| \cos(y_i, x_i) \leq \|y_i'\| \|x_i\| \cos(y_i', x_i) = \langle y_i', x_i \rangle.$$

Summing over all $i$ gives $\langle y, x \rangle \leq \langle y', x \rangle$. Using the equality of global norms, we obtain

$$\cos(y, x) = \frac{\langle y, x \rangle}{\|y\| \|x\|} \leq \frac{\langle y', x \rangle}{\|y'\| \|x\|} = \cos(y', x).$$

Thus $\cos(y, x) \leq \cos(y', x)$, as claimed. $\square$

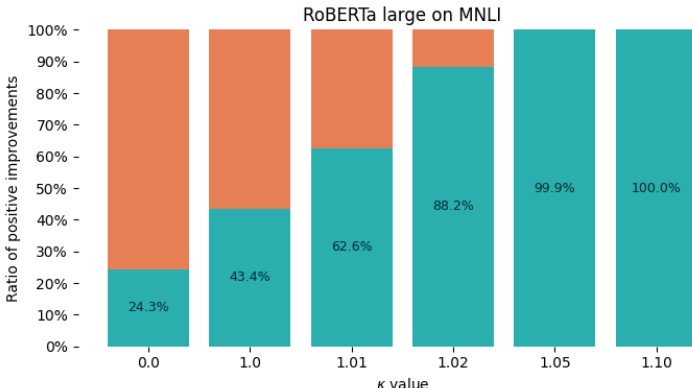

Figure 9: Effect of different values of $\kappa$ on the per-layer improvements when training RoBERTa large model on MNLI dataset. The Setting are similar to what is described in 4.

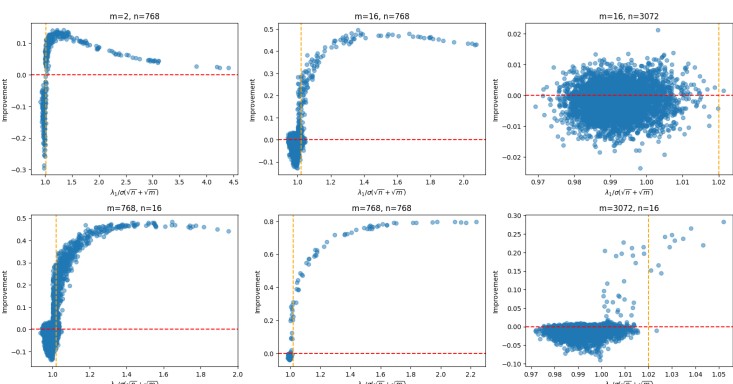

Figure 10: Scatter plot of layer improvement vs $\frac{\lambda_1}{\sigma(\sqrt{n}+\sqrt{m})}$ for different layer dimensionality. The vertical yellow line shows the threshold $\kappa$ we used in our experiments. We want the yellow line in a position to have lots of points on top right side, and few points on the bottom right side (and preferably few on top left side).

### D.2 ADDITIONAL NORM CORRECTION ABALATION RESULTS

For the sake of completeness, we calarify what we mean by the abalation. Instead of the equation 9, we use the denoised gradient without the norm correction, i.e.,

$$\text{Denoise}_{\text{unscaled}}(\tilde{g}\,|\,W) = \begin{cases} \tilde{g}\,|\,W, & \text{if } \lambda_1(\tilde{g}\,|\,W) < \kappa\,\sigma(\sqrt{n}+\sqrt{m}) \\ \hat{g}_{\text{optimal}}, & \text{otherwise} \end{cases}$$

Other than the 3, we also have the results for training both RoBERTa base and large models on MNLI dataset, as well as RoBERTa large model on SST-2 dataset. These results are presented in figures 6. It is evident from these results that without the norm correction, the improvement is not consistent, and can even be negative in some cases. Also, it is evident that with the norm correction, the improvement is consistently positive. This shows the importance of the norm correction step in our denoising function.

## E  ADDITIONAL LANGUAGE GENERATION EXPERIMENTS

In this appendix, we present additional experiments on language generation tasks, specifically on the E2E benchmark (Novikova et al., 2017) and DART (Nan et al., 2021) datasets. We used the

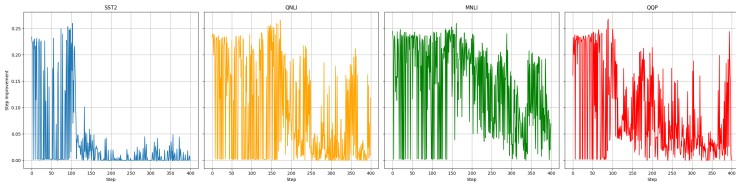

Figure 11: Improvement in cosine similarity between denoised and noisy gradients with respect to clipped gradients over training steps for different datasets. The positive values indicate that the denoising method consistently enhances the alignment between the noisy and clipped gradients throughout the training process.

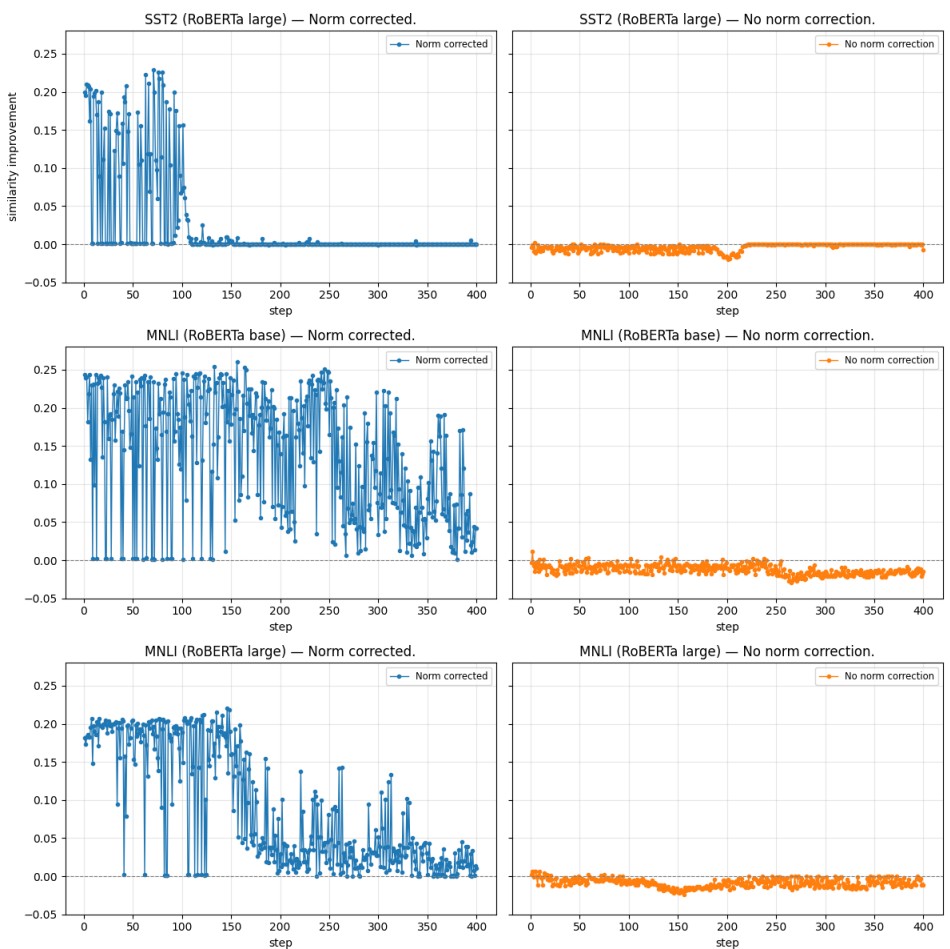

Figure 12: Comparison of whole gradient improvement when using norm correction and not using it. The Settings are similar to what is described in 4. It is evident that using norm correction results in more consistent improvement.

models from Qwen3 (Team, 2025) and Llama3.2 (Grattafiori et al., 2024) family. For training these models, we used LoRA (Hu et al., 2022) with rank 32. We used a learning rate of $2e - 4$, batch size of 64, and clipping norm of $1.0$. The privacy budget $\epsilon$ was set to $5.4$ for all experiments, and the noise multiplier was calculated using the privacy accountant from Gopi et al. (2021) to ensure a total privacy loss of $5.4$ at $400$ training steps. For generating text during evaluation, we used the same setting as in Yu et al. (2021). For the denoising, we used the same $\kappa = 1.02$ as in all other experiments, showing the robustness of our method to this hyperparameter. Similar to experiments on the GLUE benchmark datasets, we see significant improvements in performance

after a limited number of training steps. Here, we present the results after 50 training steps to highlight the sample efficiency of our denoising method. In table 4, we can see that out of the 50 different model/dataset/metric combinations, the denoised models outperformed the baseline in 49 cases, demonstrating the effectiveness of our denoising method in enhancing model performance on language generation tasks under differential privacy constraints.

The final performance of the models after 400 steps are reported in table 5. In the metrics after 400 steps, out of the 50 different model/dataset/metric combinations, the denoised models outperformed the baseline in 35 cases, showing that our method not only improves sample efficiency but also leads to better final performance in many scenarios. This, however, is still an area for further investigation to understand the cases where the baseline outperforms the denoised models at the later steps.

| Dataset | Model | Size | Variant | BLEU | ROUGE-L | METEOR | NIST | CIDEr |
|---------|-------|------|---------|------|---------|--------|------|-------|
| E2E | Qwen | 0.6B | baseline | 23.17 | 46.78 | 0.509 | 2.34 | 0.67 |
|     |      |      | denoised | **37.35** | **53.52** | **0.628** | **4.60** | **1.11** |
| E2E | Qwen | 1.7B | baseline | 36.01 | 53.57 | 0.609 | 4.64 | 1.08 |
|     |      |      | denoised | **36.95** | **54.42** | **0.624** | **4.71** | **1.14** |
| E2E | Qwen | 4B | baseline | 40.00 | 55.19 | 0.659 | 4.83 | 1.27 |
|     |      |      | denoised | **40.88** | **55.74** | **0.663** | **5.03** | **1.36** |
| E2E | Llama | 1B | baseline | 22.15 | 44.24 | 0.476 | 2.59 | 0.47 |
|     |      |      | denoised | **36.33** | **52.86** | **0.626** | **4.62** | **1.13** |
| E2E | Llama | 3B | baseline | 10.96 | 25.22 | 0.268 | 0.93 | 0.22 |
|     |      |      | denoised | **26.33** | **44.36** | **0.516** | **3.48** | **0.64** |
| DART | Qwen | 0.6B | baseline | 14.66 | 33.44 | 0.323 | 0.91 | 0.59 |
|      |      |      | denoised | **23.58** | **46.46** | **0.462** | **2.87** | **0.84** |
| DART | Qwen | 1.7B | baseline | 21.77 | 46.98 | 0.483 | 4.23 | 0.86 |
|      |      |      | denoised | **33.29** | **51.95** | **0.579** | **5.26** | **1.26** |
| DART | Qwen | 4B | baseline | 21.74 | 44.47 | **0.522** | 3.84 | 0.86 |
|      |      |      | denoised | **21.78** | **46.98** | 0.484 | **4.22** | **0.86** |
| DART | Llama | 1B | baseline | 8.88 | 37.38 | 0.368 | 2.86 | 0.41 |
|      |      |      | denoised | **13.81** | **43.12** | **0.454** | **3.68** | **0.73** |
| DART | Llama | 3B | baseline | 6.12 | 30.08 | 0.319 | 2.13 | 0.26 |
|      |      |      | denoised | **9.47** | **37.33** | **0.375** | **2.95** | **0.47** |

Table 4: Comparison of baseline vs. denoised models on E2E and DART datasets after 50 steps. Bold indicates the better value within each pair.

## F    USE OF LLMS

We have utilized large language models (LLMs) to assist in editing and refining the manuscript. LLMs were used to improve the clarity, coherence, and overall quality of the writing, ensuring that the content is presented in a clear and accessible manner.

| Dataset | Model | Size | Variant | BLEU | ROUGE-L | METEOR | NIST | CIDEr |
|---------|-------|------|---------|------|---------|--------|------|-------|
| E2E | Qwen | 0.6B | baseline | 38.77 | 53.57 | 0.661 | 4.83 | 1.17 |
|  |  |  | denoised | **38.94** | **55.07** | **0.668** | **4.87** | **1.20** |
| E2E | Qwen | 1.7B | baseline | 40.00 | 55.70 | 0.661 | 4.91 | 1.22 |
|  |  |  | denoised | **40.05** | **55.73** | **0.670** | **4.92** | **1.23** |
| E2E | Qwen | 4B | baseline | **40.58** | **56.47** | **0.678** | **5.00** | **1.32** |
|  |  |  | denoised | 40.44 | 56.33 | 0.676 | 4.99 | 1.28 |
| E2E | Llama | 1B | baseline | 37.06 | 54.94 | 0.629 | 4.65 | 1.21 |
|  |  |  | denoised | **39.70** | **55.29** | **0.662** | **4.93** | **1.30** |
| E2E | Llama | 3B | baseline | 39.02 | 54.68 | 0.659 | 4.89 | 1.25 |
|  |  |  | denoised | **39.64** | **54.83** | **0.682** | **4.91** | **1.29** |
| DART | Qwen | 0.6B | baseline | **25.99** | **50.44** | **0.595** | **3.38** | 1.01 |
|  |  |  | denoised | 25.43 | 50.23 | 0.510 | 3.37 | **1.02** |
| DART | Qwen | 1.7B | baseline | **31.07** | 52.63 | 0.572 | 4.77 | 1.26 |
|  |  |  | denoised | 30.98 | **52.89** | **0.581** | **4.93** | 1.26 |
| DART | Qwen | 4B | baseline | 29.70 | 52.22 | 0.571 | 4.63 | 1.22 |
|  |  |  | denoised | **34.82** | **54.71** | **0.594** | **5.29** | **1.38** |
| DART | Llama | 1B | baseline | **20.03** | 46.40 | 0.466 | **3.91** | **0.79** |
|  |  |  | denoised | 19.41 | **46.85** | **0.479** | 3.95 | 0.81 |
| DART | Llama | 3B | baseline | **23.56** | **48.01** | **0.512** | **4.60** | **1.00** |
|  |  |  | denoised | 21.51 | 44.25 | 0.461 | 3.80 | 0.87 |

Table 5: Comparison of baseline vs. denoised models on E2E and DART datasets. Bold indicates the better value within each pair.

