# OpenReview forum: "Sample-Efficient Differentially Private Fine-Tuning via Gradient Matrix Denoising"
_ICLR.cc/2026/Conference — Submitted to ICLR 2026_

### Official Review · Reviewer_fZ14 · 2025-10-27

**Soundness:** 3
**Presentation:** 3
**Contribution:** 2
**Rating:** 4
**Confidence:** 3

**Summary:**

This paper identifies that DP noise disrupts the inherent low-rank structure of gradient matrices. They propose a post-processing method that uses Random Matrix Theory to denoise the gradients on a per-layer basis. The core idea is to identify and reconstruct the signal singular values that emerge from the noise. The authors' experiments on RoBERTa models show that this method can achieve target validation accuracies in significantly fewer training steps, although it does not always achieve the best final accuracy.

**Strengths:**

The paper is well-written, and the proposed method is described with sufficient detail to be easily followed and understood. Besides, the authors provide a practical and honest analysis of their method's performance.

**Weaknesses:**

W1: The method's effectiveness is doubtful due to theoretical contradictions, such as the "Norm Correction" (Step C) deviating from the RMT optimal estimate (Step B). This correction provides an inflated gradient magnitude that could harm optimizers, and the method's reliance on an empirically-tuned $\kappa$ (Step A) further questions its robustness.

W2: The paper claims improved "sample efficiency" but ignores the massive computational overhead of performing SVD on every linear layer at every step.

**Questions:**

Q1: Could the authors provide a concrete comparison of the total training time or the average time per training step against the baseline?

Q2: How sensitive is the algorithm's final performance to the choice of the hyperparameter $\kappa$?

Q3: Could the authors provide an ablation on the performance without the $\kappa$-thresholding  and norm correction (Step C) to isolate its effects?

---

> ### Author Response · Authors · 2025-11-27
>
> We thank the reviewer for the constructive and encouraging feedback. We have implemented substantial revisions based on your comments and summarize the key changes below.
> 1. Computational Overhead
> We expanded the discussion and clarified the implementation details of the denoising step. With the optimized implementation—leveraging per-singular-value parallelism, batched SVD, and optional approximations—the computational overhead is below 1%, as shown in Table 1. We hope this addresses the concerns in w2 and q1.
> 2. Norm Correction
> We have significantly expanded our explanation of the necessity and correctness of the norm-correction step.
> Key clarifications:
> RMT denoising applies independently to each matrix; there is no theoretical extension of RMT that dictates relative scaling across multiple layers.
>
>
> Therefore, a mechanism is required to ensure that improving the per-layer alignment also translates to improving the alignment of the entire gradient vector.
>
>
> Theorem 1 now explicitly establishes this: if each component’s alignment improves and their norms are preserved, then the global alignment must also improve.
>
>
> Norm correction does not increase gradient magnitude; it ensures that the denoised gradient has exactly the same norm as the noisy DP-SGD gradient, thereby preserving optimizer stability.
>
>
> New ablations (Figures 3 and 12) demonstrate that without norm correction, global alignment does not improve.
>
>
> These changes address w1 and q3.
> 3. Threshold Multiplier κ
> We clarified the motivation for the threshold
> κ σ(n+m)\kappa\,\sigma(\sqrt{n}+\sqrt{m})κσ(n​+m​)
> by connecting it explicitly to the BBP phase transition. When the largest singular value does not exceed this quantity, the optimal estimator collapses to zero, and denoising is theoretically meaningless.
> We also added extensive κ-ablation results (Figures 2, 7–9). Across all experiments except the κ-ablation itself, we used a fixed κ = 1.02, demonstrating robustness.
> We hope these revisions satisfactorily address the remaining parts of w1, q2, and q3.
> We appreciate your insightful feedback and hope our clarifications and new experiments resolve your concerns.

---

### Official Review · Reviewer_zer2 · 2025-10-31

**Soundness:** 2
**Presentation:** 2
**Contribution:** 2
**Rating:** 2
**Confidence:** 4

**Summary:**

The paper studies private fine-tuning of LLMs using DP-SGD. Compared to the standard SGD, DP-SGD adds clipping and Gaussian noise in order to protect privacy. Both would affect the model performance and decrease training efficiency. This paper mainly addresses the part with added Gaussian noise. Motivated from random matrix theory, the paper proposes to add a post processing step after clipping and noise addition to recover original gradients from noisy observations. The new methods can reach 90% or 95% of the state-of-the-art performance with reduced training steps when fine-tuning RoBERTa on GLUE tasks.

**Strengths:**

I think adding a post processing step to denoise the gradients is a good idea. The motivation and the design of the algorithm are presented clearly. The studied topic is interesting to the community, as fine-tuning LLMs indeed poses privacy concerns. Introducing efficient algorithms for private fine-tuning of LLMs is of great importance.

**Weaknesses:**

1. SVD is always required for the denoising step to compute all singular vectors and singular values of gradient matrices. This adds significant amout of computational overhead. Although the paper claims on sample efficiency, computational efficiency is also or even more important in many cases. What is the total running time to reach the claimed performance? If per-step cost is significantly higher, it is unfair and potentially misleading to only compare number of steps.

2. It turns out random matrix theory provides good solutions but mainly works for the asymptotic case. There are lots of approximations and additional fixes to make it work in pratical non-asymptotic regime. I doubt whether the denoise function used in the paper is still optimal. The fixes by $\kappa$ and norm rescaling also look artificial and fragile to me. These fixes make the method seem less like a clean solution but more like a heuristic that requires careful tuning and may not be robust if a different model and a different tasks are used. Moreover, should the noise in $\kappa \sigma (\sqrt{n}+\sqrt{m})$ be $\sigma C/|B|$? Is this a typo or a wrong algorithm is used?

3. I do not think the goal in the experiment section to match 95% of SOTA with less steps makes sense. First, as said in 1, running time should also be considered. Second, in my opinion, the more important problem to be addressed in DP fine-tuning is the reduced performance due to privacy noise. Designing algorithms that achieves better trade-off between utility and privacy, i.e., better performance under the same privacy budget, is more interesting. Can the proposed algorithm reach or surpass the performance of SOTA with the same number of iterations or with the same training time?

4. The experiments are very limited. Only 4 tasks are considered on a single RoBERTa family. It is unclear whether the algorithms also work for other model family, other tasks, and scale to larger models.

**Questions:**

1. Are Figures 1 and 2 required to be this large? More interesting results can be presented if this space is saved.

2. The notation on Gaussian noise is not consistent and gives lots of confusion. In line 145 and eq. (2), it is $N(0, \sigma C)$ but $N(0, \sigma^2 C^2)$ in many other places.

3. What is the purpose of Section 3.1? Is Improvement(t) ever used? This metric is not private and could leak sensitive information.

4. Improvement in cosine similarity does not directly imply the algorithm should perform better. One reason is that there is still per-sample gradient clipping.

---

> ### Author Response · Authors · 2025-11-27
>
> We appreciate the reviewer’s time and acknowledge the positive comments about the motivation and clarity of the method. We have made extensive revisions addressing each concern. Our responses follow the reviewer’s structure.
> W1. Computational Overhead
> We added a full overhead analysis and Table 1. With the optimized implementation, overhead is <1%, and in all experiments we report both step counts and runtime.
> W4. Limited Experiments
> We expanded the study from 4 tasks / 1 model family to 6 tasks / 3 model families / 7 models—increasing model–dataset pairs from 8 to 18—and show improvements, all using the same fixed κ.
> W3. Focus on Sample Efficiency vs Final Accuracy
> We now explicitly analyze convergence speed, supported by new Figures 1 and 6. These show that noise—not clipping—is the primary factor causing >4× slower convergence. Thus, improving sample efficiency is an important and distinct objective.
> Q1. Figure Size
> We moved Figures 1–2 to the appendix to make space for ablations and sensitivity analyses.
> Q2. Noise Notation
> We corrected two mismatches in notation.
> Q3. Use of the Improvement Metric
> We clarified that Improvement(t) is not used in the algorithm (to preserve DP). It is used only for evaluation and analysis. Its purpose is now more clearly explained.
> Q4. Improvement in Cosine Similarity vs Performance
> We added discussion explaining that cosine improvement does not ensure performance improvement by itself, but our ablations show that alignment increases correspond to faster convergence across tasks and models.
> W2. Detailed Technical Response
> We reorganized and clarified Section 3.2, added more theoretical explanation, and provided additional ablations on norm correction and thresholding. We also confirm that:
> The κ⋅σ(√n + √m) expression is correct and aligns with the BBP threshold.
>
>
> We could not locate the expression “σ*C/|B|” in our manuscript and kindly ask for clarification.
> We have saved the answer for W2 for last, as it is more detailed.
>
> 2. Norm Correction
> We have significantly expanded our explanation of the necessity and correctness of the norm-correction step.
> Key clarifications:
> RMT denoising applies independently to each matrix; there is no theoretical extension of RMT that dictates relative scaling across multiple layers.
>
>
> Therefore, a mechanism is required to ensure that improving the per-layer alignment also translates to improving the alignment of the entire gradient vector.
>
>
> Theorem 1 now explicitly establishes this: if each component’s alignment improves and their norms are preserved, then the global alignment must also improve.
>
>
> Norm correction does not increase gradient magnitude; it ensures that the denoised gradient has exactly the same norm as the noisy DP-SGD gradient, thereby preserving optimizer stability.
>
>
> New ablations (Figures 3 and 12) demonstrate that without norm correction, global alignment does not improve.
>
>
> These changes address w1 and q3.
> 3. Threshold Multiplier κ
> We clarified the motivation for the threshold
> κ σ(n+m)\kappa\,\sigma(\sqrt{n}+\sqrt{m})κσ(n​+m​)
> by connecting it explicitly to the BBP phase transition. When the largest singular value does not exceed this quantity, the optimal estimator collapses to zero, and denoising is theoretically meaningless.
> We also added extensive κ-ablation results (Figures 2, 7–9). Across all experiments except the κ-ablation itself, we used a fixed κ = 1.02, demonstrating robustness.
> We hope these revisions satisfactorily address the remaining parts of w1, q2, and q3.
> We appreciate your insightful feedback and hope our clarifications and new experiments resolve your concerns.

---

### Official Review · Reviewer_erjr · 2025-11-02

**Soundness:** 3
**Presentation:** 2
**Contribution:** 3
**Rating:** 6
**Confidence:** 2

**Summary:**

This paper proposes a post-processing denoising method for DP fine-tuning of LLMs using DP-SGD. By applying random matrix theory-based denoising (singular value shrinkage and norm correction) to the noisy gradients of linear layers, the method aims to restore low-rank structure and improve sample efficiency. The approach is evaluated on GLUE benchmarks with RoBERTa models, showing consistent reductions in the number of training steps needed to reach target accuracy under DP constraints. The method leverages the post-processing invariance of DP to preserve privacy guarantees.

**Strengths:**

-	This paper addresses an important and practical challenge: improving sample efficiency in DP fine-tuning of LLMs.
-	The authors applies theoretically motivated random matrix denoising to the DP-SGD context, with clear exposition and solid background. The method is modular, simple to implement, and preserves DP guarantees via post-processing.
-	Empirical results on GLUE tasks and RoBERTa models show consistent speedup in convergence (steps to reach target accuracy).

**Weaknesses:**

-	The method does not always improve final accuracy; in some cases, the baseline outperforms the proposed approach, and this is not analyzed in depth.
-	The threshold hyperparameter (κ) is tuned on a single dataset; robustness and generalization are not validated.
-	The method only addresses noise-induced degradation, not the often more severe impact of gradient clipping in DP-SGD.

**Questions:**

- How does the method compare to alternative DP optimization or post-processing strategies?
- How robust is the method to the choice of threshold κ and norm correction? Did you tune these on each task/model or fix them universally?
- How does the method compare to alternative DP optimization or post-processing strategies?

---

> ### Author Response · Authors · 2025-11-27
>
> We thank the reviewer for a thoughtful and positive evaluation. We address each of your points below, now explicitly referencing the new figures added in the revised submission.
> Final Accuracy
> While the proposed denoising method consistently improves sample efficiency, it does not always improve final accuracy. We clarified this distinction in the paper and added discussion in Section 4.
>  To illustrate this, we now explicitly refer to the new convergence plots (Figure 1 and Figure 6 in the updated manuscript
> new_submission
> ), which show:
> denoising reaches target accuracy much sooner,
>
>
> but after very long training, the baseline may catch up.
>
>
> This directly addresses the reviewer’s concern.
> κ Robustness
> We added four new κ-ablation studies:
> Figure 2 (main paper)
>
>
> Figures 7, 8, and 9 (appendix)
>
>
> These demonstrate that κ controls the fraction of layers that meaningfully benefit from denoising, but overall performance trends remain similar across a wide κ range.
>  As noted in the paper, κ was tuned on a single model–dataset pair and then fixed for all remaining 17 model–dataset combinations.
> Norm Correction and Its Impact
> We also added:
> Figure 3 (main paper)
>
>
> Figure 12 (appendix)
>
>
> These support the necessity of the norm-correction step by showing that without it, global gradient alignment does not improve, even when per-layer alignment does.
> Comparison to Other DP Optimization Strategies
> Q1 and Q3. While there has been previous works on post-processing, they either have been impractical for big models [1], or can be applied alongside our denoising method and may be orthogonal to it [2].  While the latter is an interesting research question, it may be more proper to attach it in a different research.
>
>
>
>
>
> [1] Borja Balle and Yu-Xiang Wang. Improving the Gaussian mechanism for differential privacy:
> Analytical calibration and optimal denoising. In Jennifer Dy and Andreas Krause (eds.), Pro-
> ceedings of the 35th International Conference on Machine Learning, volume 80 of Proceed-
> ings of Machine Learning Research, pp. 394–403. PMLR, 10–15 Jul 2018. URL https:
> //proceedings.mlr.press/v80/balle18a.html.
>
> [2] Ziawei Zhao, Zhenyu Zhang, Beidi Chen, Zhangyang Wang, Anima Anandkumar, and Yuandong
> Tian. Galore: Memory-efficient llm training by gradient low-rank projection. arXiv preprint

---

### Meta-Review · Area_Chair_7UWa · 2026-01-14

**Summary:**

The paper proposes a gradient denoising algorithm that aims at improving the noisy gradients in DP-SGD. Reviewers find the idea of using post-processing. However, there were concerns about the denoiser's effectiveness. Specifically, the denoiser can only reduce the number of samples needed to achieve 95% of the best private accuracy. The main challenge in privacy preserving ML is loss of accuracy, not the sample complexity. There are also no convincing justifications for why the algorithm only works for low sample complexity. I would recommend rejecting the paper for this reason.

**Reviewer Concerns:**

Reviewers erjr and zer2 had concerns about the significance of improving sample complexity compared to overall accuracy. I don't think this comment is adequately addressed.

Reviewers had concerns about total computational cost of the denoising method. Reviewers adequately address that by optimizing the denoising overhead to be roughly 1% of the total cost.

Reviewer zer2 asked for more experiments on more datasets and models and authors expanded their set of experiments.

**Reviewer Scores:**

Reviewer zer2 may increase their score because of the new experiments.

---

### Decision · Program_Chairs · 2026-01-26

Reject